# The Role of Bronchoalveolar Lavage in Systemic Sclerosis Interstitial Lung Disease: A Systematic Literature Review

**DOI:** 10.3390/ph15121584

**Published:** 2022-12-19

**Authors:** Martina Orlandi, Laura Antonia Meliante, Arianna Damiani, Lorenzo Tofani, Cosimo Bruni, Serena Guiducci, Marco Matucci-Cerinic, Silvia Bellando-Randone, Sara Tomassetti

**Affiliations:** 1Scleroderma Unit, Department of Experimental and Clinical Medicine, Division of Rheumatology AOUC, University of Florence, 50134 Florence, Italy; 2Unit of Immunology, Rheumatology, Allergy and Rare Diseases (UnIRAR), IRCCS San Raffaele Hospital, 20132 Milan, Italy; 3Department of Clinical and Experimental Medicine, University of Florence, and Division of Interventional Pulmonology, Careggi University Hospital, 50134 Florence, Italy

**Keywords:** systemic sclerosis, scleroderma, interstitial lung disease, Bronchoalveolar Lavage, alveolitis

## Abstract

The role of Bronchoalveolar Lavage (BAL) in the evaluation of systemic sclerosis (SSc) interstitial lung disease (ILD) is still controversial. The aim of this systematic literature review was to investigate the use of BAL in SSc-ILD, and to focus on the pros and cons of its real-life application. Methods: PubMed, Cochrane, and Embase were questioned from inception until 31 December 2021. Results: Eighteen papers were finally analyzed. A positive correlation was observed between lung function and BAL cytology; in particular, BAL neutrophilia/granulocytosis was related to lower diffusing capacity for carbon monoxide (DLCO) values and lower forced vital capacity (FVC). Moreover, a positive correlation between BAL cellularity and high-resolution computed tomography (HRCT) findings has been reported by several authors. Cytokines, chemokines, growth factors, coagulation factors, and eicosanoids have all been shown to be present, more often and in higher quantities in SSc-ILD patients than in the health control and, in some cases, they were related to more severe pulmonary disease. There was no consensus regarding the role of BAL cellularity as a predictor of mortality.

## 1. Introduction

Systemic sclerosis (SSc) is characterized by fibrosis of the skin and internal organs, vasculopathy, and immune dysregulation [1]. Interstitial lung disease (ILD) is one of the most frequent complications and the main cause of death [2]. In approximately 75% of SSc patients, the radiological aspects of SSc-ILD are represented by the non-specific interstitial pneumonia (NSIP) pattern, which is usually characterized by ground glass opacities (GGO) and reticulations (RET). The relatively less frequent pattern is the usual interstitial pneumonia (UIP) [3] in which the predominant feature is honeycombing (HC) [4,5,6,7,8]. High-resolution computed tomography (HRCT) is the gold standard for the diagnosis of SSc-ILD and can also provide important information on the ILD extension and patient’s prognosis [4,9]. Pulmonary function tests (PFT) are also fundamental for the evaluation of disease severity at baseline, and of disease progression at follow-up [10]. Bronchoalveolar lavage (BAL) is performed as part of bronchoscopy and consists of the instillation and subsequent withdrawal of sterile saline [11]. It investigates the lower respiratory tract through the sampling of cellular and acellular (biochemical) components from alveolar spaces. In healthy subjects, 90–95% of BAL cells consist of alveolar macrophages, and the leukocyte component does not usually exceed 3% for neutrophils, 2% for eosinophils, and 15% for lymphocytes [12,13]. An abnormally high number of BAL leukocytes, in particular, granulocytes (neutrophils and eosinophils) or lymphocytes, is often referred to as “alveolitis”, since it is believed to reflect an inflammatory process in ILD patients. Some contributions report that the presence of alveolitis may play a crucial role in the clinical progression of alveolitis in SSc patients [14,15,16,17,18]. This implies that the characterization of alveolitis by BAL could be clinically useful for the management of SSc-ILD patients. However, the role of BAL in ILD is still controversial. Moreover, the invasiveness and the technical differences in the procedure and in the processing of BAL fluid have limited the use of this technique in clinical practice [13,18,19].

The aim of our study was to investigate the use of BAL and its different aspects in SSc-ILD through a systematic literature review (SLR), to focus on the pros and cons of its real-life application.

## 2. Materials and Methods

We conducted an SLR to investigate the role of BAL in SSc-ILD patients; the research of publications of interest was conducted by questioning PubMed, Embase, and Cochrane databases. Each database was searched from the database inception date until 31 December 2021. Search terms included a combination of database-specific controlled vocabulary terms and free-text terms relating to BAL and SSc-ILD diagnostic and prognosis. PubMed and Cochrane were questioned for “bronchoalveolar lavage AND (scleroderma OR systemic sclerosis) AND interstitial lung disease AND (diagnosis OR diagnostic OR prognosis OR prognostic)”. Embase was questioned for “‘bronchoalveolar lavage’: ti,ab AND (scleroderma: ti,ab OR ‘systemic sclerosis’:ti,ab) AND ‘interstitial lung disease’:ti,ab AND diagnos*:ti,ab OR (‘lung lavage’/exp AND ‘scleroderma’/exp AND ‘interstitial lung disease’/exp AND (‘diagnosis’ OR ‘diagnostic procedure’ OR ‘prognosis’/exp OR prognos*)”. All articles were identified by searching the three databases and were imported into Mendeley for screening. After deduplication, two screening rounds were performed, as described in Figure 1. In the first round, two reviewers (M.O. and L.M.) evaluated, in duplicate, titles and abstracts in terms of relevance for both BAL and SSc-ILD. In the second round, the full texts of the articles included during the first round were retrieved and re-assessed for eligibility. Possible discordance during study selections was discussed with a third reviewer (S.BR) to reach a consensus. Study inclusion criteria included peer-reviewed publication in English, with population-based studies that reported an association between BAL and SSc-ILD. Case reports, reviews, congress abstracts, letters to the editor, or editorials were excluded, as well as case series with less than 10 SSc patients. Additional references pertaining to the research aspects of BAL in SSc-ILD were also reviewed. A detailed flow chart describing the study inclusion and exclusion process is presented in Figure 1. Studies were selected according to the PEO (population, exposure, and outcomes) framework outlined in Appendix A. The quality assessment of diagnostic accuracy studies (QUADAS) for articles included in the systematic review is summarized in Appendix A.

Data were extracted and included in an electronic database. The database included data about study design (retrospective cohort/prospective cohort/cross-sectional/clinical trials); total number of SSc patients in the study (with and without ILD); SSc diagnosis (ACR 2013 criteria/ARA 1980); type of SSc (limited and diffuse); demographical data (age, sex); disease duration; smoke or environmental exposure; pulmonary functional test (FVC/FEV1/DLco); therapy (previous/current); HRCT (ILD extension/predominant pattern/predominant CT features; BAL (site/presence of alveolitis/cellularity); follow-up (clinical progression/functional progression/radiological progression/mortality).

### Statistical Analysis 

Statistical analyses were performed using SPSS 20.0 software (Armonk, NY, USA: IBM Corp.). The descriptive analysis was pooled according to the sample size of each manuscript. All continuous variables were described by mean ± SD, while for categorical ones, by absolute and relative frequencies.

## 3. Results

### 3.1. Study Selection

The initial search retrieved 331 results from MEDLINE, 127 from PubMed, 200 from Embase, and 4 from Cochrane Collection. Aiming to select the most relevant articles and of the highest quality, we included peer-reviewed publications in English, with population-based studies that reported an association between BAL and SSc-ILD, while case reports, reviews, congress abstracts, letters to the editor, or editorials were excluded, as well as case series with less than 10 SSc patients. In Figure 1, the flowchart with details on the included and excluded manuscripts is shown. First-stage screening by reviewing titles and abstracts excluded 212 publications, while 45 articles were identified for the second-stage screening through full-text analysis. Eventually, 18 articles met all the inclusion criteria and constituted the final pool of this SLR (Figure 1) [20,21,22,23,24,25,26,27,28,29,30,31,32,33,34,35,36,37].

### 3.2. Study Characteristics

Seven (38.9%) of the included manuscripts presented data from prospective studies [23,28,30,31,37], 5/18 (27,8%) were retrospective [20,22,27,29,34] and 6/18 (33,3%) were cross-sectional [21,26,32,33,35,36]. The studies were published between 1995 and 2012 and 11/18 studies (61,1%) were case-control reports [20,21,22,25,26,27,30,32,35,36]. The selected studies included a total of 1517 patients of which 1042 were SSc-ILD patients.

### 3.3. SSc Patients 

In most articles (88.9%), SSc patients were classified according to the American Rheumatism Association (ARA) Diagnostic and Therapeutic Criteria 1980 [38]. As expected, the study population included mostly females (mean distribution 78.7 ± 49.7%) with a mean age of 49.7 ± 11.9 years. The limited cutaneous subset population represented 63.4% of patients (*n* = 263), while 36.6% were affected by the diffuse cutaneous subset (*n* = 187) (the subset was specified in seven articles [20,22,23,25,26,34,35]). The mean disease duration was 5.4 ± 5.6 years. No data about patient’s nationality, mRSS value, and disease activity values (VES, PCR, tendon friction rubs) were available. Autoantibodies subset definition was specified only in three articles [23,34,35]; in a total of 172 patients, 80 (44%) were anti-centromere antibodies (ACA) positive, and 56 (26.7%) were scl-70 positive. The remaining were positive for other non-ACA and non-Scl 70 autoantibodies. 

Exposure was specified only for smoking: 53.5% of SSc-ILD patients were nonsmokers, former in 27.5%, and current in 12.3% of the cases. In Table 1 are the most important clinical data. 

### 3.4. Therapy

Only four studies reported data about previous therapies in SSc patients who underwent BAL, for a total of 66 patients [22,23,34]. Among them, the most common previous therapy was corticosteroid (48 patients) [22,23,34], Cyclophosphamide (7 patients) [22,23], while Mycophenolate Mofetil (1 patient) [24] and other drugs (Azathioprine, Methotrexate, and etanercept) (10 patients) were less frequent [22,24]. Six papers specified the current therapy [20,23,24,25,27,34], for a total of 339 patients. The most common current medication was cyclophosphamide (162 patients) [20,23,25,34] and corticosteroids (119 patients) [20,24,28]. There was 1 patient on Mycophenolate mofetil [24] and 57 on other drugs (azathioprine was the most reported, otherwise not specified) [20,23,24,28].

### 3.5. Pulmonary Function Tests (PFT)

The PFT data at baseline were available for 84.5% of SSc-ILD patients [20,22,23,24,27,28,29,30,31,34,35,37]. The mean forced vital capacity (FVC) value was 79.3 ± 19.7 (mean ± SD); Forced Expiratory Volume in the 1st second (FEV1) 72.8 ± 16.6 (mean ± SD) and diffusing capacity for carbon monoxide (DLCO) 54.8 ± 16.9 (mean ± SD). 

### 3.6. HRCT Imaging 

Despite HRCT being performed on 99.2% of patients included in the evaluated studies, data about CT were reported only in a half of the studies [20,21,23,24,25,26,30,35,37]. The extension of lung involvement was evaluated only in two studies [24,26]. Detailed data about HRCT patterns were reported only in 4/18 manuscripts (for a total of 262 patients) [25,26,35,36]: GGO were present in 208 SSc patients [25,26,35,36], fibrosis in 167 patients [21,25,35], HC in 101 patients [23,25], and RET in 28 patients [35,36].

Detailed data about HRCT patterns were reported in one paper only [20]: the NSIP pattern was present in 62 patients, while the UIP pattern was found in 6 patients.

### 3.7. BAL

BAL was performed in 92.5% of SSc patients, but no details regarding the technique were available. In 10 papers (55,5%), for a total of 397 patients, the site of BAL execution was indicated [21,22,25,27,28,31,32,33,35,37], being the right middle lobe in 50% of the papers (5/10) [25,27,28,31,32] in a total of 260 patients (65.5% of patients). In 122 patients, it was the right middle and lower lobes and lingula [21,33,37]. In one study, the mid-lung zones were investigated (usually in the right middle lobe or lingula; although, the superior segment of the lower lobe was chosen in two cases) and a second segment in the basilar portion of the lower lobe [22]. In one study, BAL was performed in the most affected lobe identified at HRCT [35].

The data on BAL cellularity were available from nine studies (50%) (309 patients) [21,23,27,28,29,30,31,35,37]. In the analyzed studies, BAL cellularity was represented on average by 6.3 ± 9.5% of neutrophils, 11.1 ± 9.4% of lymphocytes, 62.5 ± 13.4% of macrophages, and 1.7 ± 2.1% of eosinophils. Significant lymphocytosis (>15%) was reported in four out of nine studies, reflecting two different BAL phenotypes of SSc-ILD: with and without increased lymphocytes (as detailed in Table 2). The lymphocyte sub-typing was described only in two papers [23,31]. The average value of CD4 was 43.3 ± 14%, and the average value of CD8 was 32.7 ± 14.1% (mean ± SD). The molecular profile was heterogeneously investigated in four studies [21,26,27,30]. The following parameters were analyzed: Elastase/apha1-antitrypsin [21], collagenase activity [21], CCL18 [30], B-thromboglobulin (BTG) [27], platelet factor 4 (PF4) [27], serum SP-D [26], and serum KL-6 [26]. Detailed data are reported in Table 3.

### 3.8. Prognosis

Only 9/18 papers had radiological/functional or clinical data about follow-up [20,23,24,25,30,31,32,34].

The mean duration of follow-up was 59 months. Most of the studies reported a functional, clinical, or radiological decline: only radiological [25], only functional [30,34], clinical and functional [20], functional and radiological [23,32], and functional clinical and radiological [24]. Disease progression was determined by either PFT [20,23,24,30,32,34], HRCT [23,24,25,26,32], clinical [24] decline, or death [20,24,34].

### 3.9. Subanalysis of Data from Patients with Alveolitis on BAL

In BAL, alveolitis was investigated in six studies [22,23,25,26,32,34], and detailed in five [21,23,26,34,37]. Two studies were prospective [23,37], two were cross-sectional [21,26], and one was retrospective [34]. It was interesting to observe that the definition of alveolitis was quite heterogeneous among papers. De Santis et al. and Volpinari et al. defined alveolitis when the percentage of neutrophils was >3% and/or eosinophils > 2% [34], while Behr et al. defined an active BAL with polymorphonuclear leukocytes > 5% and/or lymphocytes > 15% [37]. Hant et al. defined alveolitis by either BAL or HRCT by the presence of any GGO and/or right middle lobe BAL that revealed ≥3.0% neutrophils and/or ≥2.0% eosinophils when a minimum of 400 cells were counted [22].

The total number of SSc-ILD patients included in the alveolitis group was 180. In most articles (88.9%), SSc was classified according to ARA 1980 criteria [38]. As expected, the study population included mostly females (79.4%) with a mean age of 52.5 ± 10.1 years (mean ± SD). A total of 61.8% of patients were affected by lcSSc and 38.2% by dcSSc. The disease duration was of 6.7 ± 7.0 years. Anti-Scl 70 were positive in 60.3% while ACA were positive in 8.8% of patients. The mean FVC value was 87.2 ± 24.4 (mean ± SD) and DLCO 56.1 ± 14.3. BAL cellularity was represented by 10.7 ± 6 of neutrophils, 5.9 ± 4.9% of lymphocytes, 81.3 ± 10.8 (mean ± SD) of macrophages, and 1.8 ± 1.9 (mean ± SD) of eosinophils. Only one study investigated the presence of CD4 and CD8 in the BAL of SSc-ILD patients with alveolitis: the mean values of CD4 and CD8 were 43.9 ± 13.9% and 34.1 ± 15%, respectively. Detailed data are available in Table 2.

## 4. Discussion

Over the last 20 years, numerous studies have focused on the role of the BAL analysis in the assessment of SSc-ILD. The BAL provides cells and acellular components from the bronchoalveolar units and, therefore, might be considered a useful technique to gain insight into the pathophysiology of SSc-ILD patients. However, our SLR shows that the studies are very heterogeneous in terms of study design, end points, investigated clinical variables, enrollment, disease activity, previous or current therapies, and HRCT pattern. Moreover, the BAL procedure and the acellular component analysis, as well as the interpretation of the results, are significantly different. Clearly, one of the major limits of BAL is the technical aspect, which significantly biases the reproducibility and the interpretation of the results. Indeed, BAL invasiveness limits its use and restricts its performance to specialized centers, even if the procedure is overall easy and safe.

### 4.1. BAL Method

Several technical factors regarding BAL execution and/or processing (the site from which BAL is performed and the amount of fluid used for BAL) still remain a matter of discussion. No sufficiently clear data regarding this issue were provided from the selected studies. In the majority of studies (five), the right middle lobe was the main site of the lavage (50%). Three studies performed the lavage in the right middle lobe, lingula, and lower lobes. The lower lobe lavage was mainly due to the predominant localization of SSc-ILD. In one paper [35], BAL was performed in the most affected lung region identified by HRCT (HRCT-guided BAL). The link between the site of lavage and the cytological analysis of BAL fluid in SSc was also investigated. Clements et al. [22] reported that the analysis of lavages taken from two separate sites was more frequently associated with a diagnosis of alveolitis. Moreover, the middle lobe and the right lower lobe are easily accessible to microorganisms as a consequence of aspiration and are frequent sites of pneumonia and bronchiectasis, as is also found in the general population [39]. Whether unexpected infections may contribute to a more frequent diagnosis of alveolitis in BAL obtained from these lobes remains to be established. In fact, as known, there are several reasons for a higher risk of infection in patients with SSc: the dysregulation of the autoimmune system, the disruption of local lung architecture due to fibrosis (for example bronchiectasis), systemic immunosuppression, and aspiration as a consequence of gastroesophageal reflux. In this context, BAL is considered the best method to obtain specimens from the lower respiratory tract for the identification of lung infection, especially in the immunocompromised patient with pulmonary infiltrates.

### 4.2. Patient’s Related Factors Could Influence BAL Results

Most of the patients included in our SLR were affected by lcSSc; according to Goldin JG et al. [25], non-significant differences between lcSSc and dcSSc patients were found for what concerns BAL findings. The mean disease duration in the selected studies was of 5.4 years and this could indicate that BAL is usually performed in the early phase of the disease to obtain some diagnostic and prognostic information. Patient-related factors may also affect the results of BAL cytological findings including environmental exposure, smoking habits, the presence of other lung diseases, the presence of respiratory tract infections, and aspiration of stomach contents, as well as the influence of drugs. Only smoking habits were investigated in the selected articles, which showed that 53% of SSc-ILD patients were nonsmokers.

### 4.3. The Role of Alveolitis as a Mirror of Lung Inflammation in ILD

An increased percentage of neutrophils, eosinophils, and/or lymphocytes in BAL define an “alveolitis”, and it is believed to mirror an inflammatory process of the lower respiratory tract (see Table 1). The concept of alveolitis is important because inflammation usually precedes and may lead to lung fibrosis in SSc-ILD pathogenesis [38]. However, the utility of BAL to detect “subclinical alveolitis” has not been shown to add value in predicting subsequent clinically significant disease [18]. Although the criteria for alveolitis varied among the studies, BAL granulocytosis was a consistent part of the definition [22,23,25,26,32,34]. An elevated percentage of neutrophils defined alveolitis in all studies, but the cut-off value varied between 3% and 5% [22,23,25,26,32,34]. Another common criterion for alveolitis is an increased number of eosinophils, with cut-off values of 2% in our series. Lymphocyte percentage was also considered for an alveolitis definition in Behr et al.’s study [37], where the cut-off was 15%.

The role of alveolitis as a mirror of lung inflammation in ILD is still debated. Alveolitis is associated with more severe lung impairment as defined by lung function tests and overall lung function.

Regarding HRCT scores, there is poor evidence at this time to recommend BAL cytology as an independent predictor of outcome in SSc-ILD. Bouros et al. [20] showed that in SSc-ILD, neutrophilia in BAL reflects more extensive lung disease on HRCT scan, rather than being an independent predictor of outcome. They demonstrated the correlation between neutrophilia and/or eosinophilia on BAL, while fibrosis is the predominant finding at lung biopsy [20], and fibrotic features predominate at HRCT [4]. De Santis et al. [23] reported that the presence of ILD on HRCT, regardless of the presence of alveolitis on BAL, is associated with a radiological and functional worsening of disease in half of the 73 cases. However, SSc patients with ILD at HRCT and with alveolitis at BAL had a higher risk for restrictive lung disease and HC, and to experience ILD progression at HRCT (worsening in HC score or developing HC). Moreover, they found that only SSc patients with alveolitis on BAL developed HC on HRCT after 1-year follow-up. These data supported a significant prognostic role of alveolitis. Moreover, in the 36 months follow-up, the presence of ILD progression at HRCT was associated with the presence of eosinophils, with an inverted CD4/CD8 ratio and with a higher CD19 percentage count in the BAL or with a positive BAL microbiological culture.

### 4.4. The Role of Alveolitis in ILD Prognostic Evaluation

It was postulated that BAL cellularity could predict the ILD evolution in SSc patients. Moodley et al. [28] reported a significantly higher total cell count and significantly higher neutrophils and eosinophils compared to healthy controls in the BAL fluid of SSc patients without ILD, confirming subclinical inflammation. This is in line with the previous observation by Harrison et al. [40], who showed abnormal BAL cytology and pulmonary interstitial inflammation in open-lung biopsy specimens from SSc patients without clinical ILD. Anyway, this cohort of patients without ILD was followed up for 13 months, and none developed any evidence of ILD. Therefore, the role of “subclinical alveolitis” alone to predict the development of subsequent clinically significant ILD has been discarded. Goh et al. [24] reported that neutrophilia on BAL was associated with early mortality (hazard ratio 8.40, *p* = 0.005), independent of disease severity, but not with late mortality or with time to decline in pulmonary function or progression-free survival. Neither eosinophilia nor lymphocytosis on BAL was correlated with mortality, rapidity of functional worsening, or progression-free survival. These findings were unchanged when treatment status was considered or when BAL neutrophil and eosinophil profiles were evaluated together (as the presence or absence of granulocytosis).

### 4.5. The Role of BAL Cytokine Expression

Studies have revealed that SSc-ILD patients show numerous aspects in BAL fluid that might play important roles in the inflammatory and fibrosing process underlying SSc-ILD. Cytokines, chemokines, coagulation factors, growth factors, and eicosanoids have all been found more often and in higher quantities in SSc-ILD patients than in the health control.

The cytokines analyzed are Elastase/apha1-antitrypsin, collagenase activity, CCL18, B-thromboglobulin (BTG), platelet factor 4 (PF4), serum SP-D, and serum KL-6 [26,27,28,30,31,32]. Whereas BAL remains a valuable research tool for studying the pathogenesis of SSc-ILD, the clinical utility of BAL remains unclear and requires further evaluation.

### 4.6. Correlation between BAL and Histopathological Pattern

Only one article correlates the BAL with the histopathological findings. Bouros et al. [20] found that the great majority of patients with fibrosing alveolitis on BAL have a histologic pattern of NSIP (77%), further subcategorized in cellular NSIP and fibrotic NSIP rather than UIP, end-stage lung disease or other patterns, in contrast to patients with idiopathic interstitial pneumonia. In the follow-up period, 22 deaths were registered mostly in UIP/end-stage lung disease. According to proportional hazards analysis, mortality was associated with lower baseline DLCO (*p* = 0.004) and FVC levels (*p* = 0.007).

Fibrotic and Cellular NSIP did not show significant differences in terms of survival and serial FVC and DLCO. The mortality rate in NSIP was higher in association with worse basal DLCO values (*p* = 0.04), BAL eosinophilia (*p* = 0.02), and a decline in DLCO values (*p* = 0.04) during the following 3 years. However, although a number of markers of lung disease were associated with survival, the histopathologic distinction between cellular and fibrotic NSIP had no prognostic implication, and there was an overall small functional decline in treated patients with NSIP in the 3 years after presentation. The Authors concluded that this outcome is connected more strongly to disease severity at presentation and serial DLCO trends than to histopathologic findings.

### 4.7. Correlation between BAL and HRCT Features

In the literature, some studies have addressed the association between BAL cytology and findings on chest radiographs, but we decided to exclude them from the final analysis because of the low sensitivity of chest radiographs in detecting ILD. Harrison et al. [41] previously reported that the percentage of BAL neutrophils was significantly elevated in SSc patients with and without features of ILD on HRCT, suggesting that BAL neutrophilia may be present in the earliest stages of SSc-ILD. Wells et al. [42] reported that BAL neutrophilia is usually linked to extensive fibrotic disease, whereas BAL eosinophilia is often seen in less advanced disease, when CT appearances suggest lung inflammation. In fact, they show that BAL neutrophils were markedly higher in association with massive disease on CT and that the degree of a reticular pattern on CT correlated with the neutrophil percentage count and total neutrophil count. Moreover, BAL eosinophils were increased in less extensive as well as in extensive disease when compared with lobes with a normal CT appearance; eosinophil percentage counts, but not total eosinophil counts/mL, correlated with the extent of a ground-glass pattern on CT (trough to represent inflammation). As reported before, Bouros et al. [20] showed that in SSc-ILD, neutrophilia in BAL reflects more extensive lung disease on HRCT scan, and De Santis et al. [23] reported that SSc patients with ILD at HRCT and with alveolitis at BAL had a higher risk for restrictive lung disease and HC, and to experience ILD progression at HRCT (worsening in HC score or developing HC).

Goldin et al. [25] reported a positive correlation between the presence of GGO and the increased number of acute inflammatory cells found in BAL fluid in 148 patients. BAL fluid was compatible with active alveolitis in 70.1% of randomized participants in the SLS 1 study. Of these 102 participants with positive BAL results, 55.9% had GGO. Overall, there was agreement between the CT scan findings and BAL in 59.7% of participants. A significantly moderate association was found between BAL outcomes and both GGO and HC. An insignificant correlation was detected between abnormal BAL findings and PF.

### 4.8. Correlation of BAL and Pulmonary Functional Tests

Several studies evaluated the relationship between DLCO and alveolitis BAL cytology, reporting an association. The presence of alveolitis is also associated with greater severity of restrictive ventilatory defect and lower FVC [18,37]. The role of DLCO in the evaluation of SSc-ILD patients may be highlighted: the decrease in DLCO correlates not only with the presence of neutrophilia or eosinophilia on BAL, but also with the BAL CD19 percentage count, inverted CD4/CD8 ratio, and CCL-18 concentrations. Moreover, it must be underlined that, as reported by Clements et al. [22], the duration of SSc-ILD correlates with the DLCO decrease, not paralleled by a reduction in FVC value.

## 5. Conclusions

In the literature, several studies on BAL in SSc-ILD assessment are available, mostly from the 1990s, when the treatment options and cellular–molecular investigations were significantly limited when compared to nowadays. The eighteen contributions included in this SLR are heterogeneous and do not allow a systematic analysis of data. As a result, our work suggests a future agenda that should include prospective clinical trials aiming to (1) standardize the technique of performing the BAL (place of performance, method of performance) in SSc-ILD; (2) characterize the alveolitis detected at BAL (lymphocytic, neutrophilic, eosinophilic, and mixed); (3) investigate novel BAL molecular biomarkers; (4) correlate BAL with functional, HRCT profiles and histopathological findings; and finally (5) correlate BAL findings with treatment response, disease evolution, and outcomes.

Clearly, much work still must be done to understand the real diagnostic potential as well as the prognostic role of BAL in SSc ILD.

## Figures and Tables

**Figure 1 pharmaceuticals-15-01584-f001:**
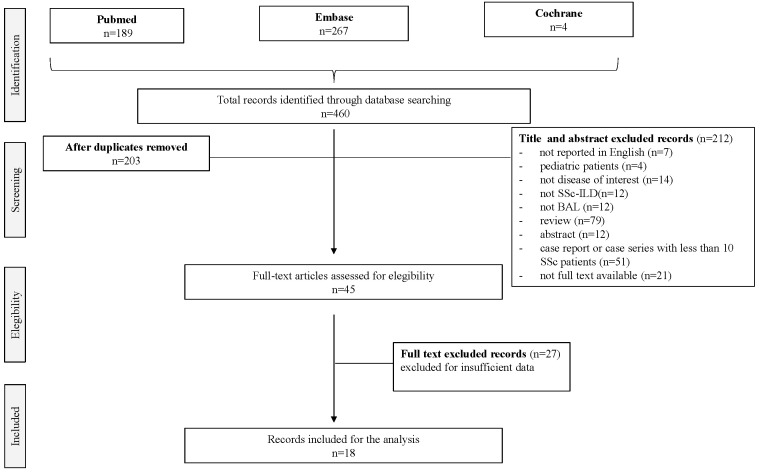
PRISMA flow chart. Flow chart describing the study inclusion and exclusion process.

**Table 1 pharmaceuticals-15-01584-t001:** Main patients, characteristic in selected studies.

	**All BAL**	**BAL with Alveolitis**
	1447 pz	186 pz
**Patients’ characteristics**		
lcSSc (%)	63.4	61.8
dcSSc (%)	36.6	38.2
Disease duration (years ± SD)	5.4 ± 5.6	6.7 ± 7.0
Female (% ± SD)	78.6	77.8
Mean age (years ± SD)	49.7 ± 11.9	52.5 ± 10.1
Never smokers (%)	53.5	
Former smokers (%)	27.5	
Current smokers (%)	12.3	
**PFT**		
FVC (pred % ± SD)	79.3 ± 19.7	87.2 ± 24.4
FEV1 (pred % ± SD)	72.8 ± 16.6	
DLCO (pred % ± SD)	54.8 ± 16.9	56.1 ± 14.3
**BALF cellularity**		
Neutrophils (% ± SD)	6.3 ± 9.5	10.7 ± 6
Lymphocytes (% ± SD)	11.1 ± 9.4	5.9 ± 4.9
Macrophages (% ± SD)	62.5	81.3 ± 10.8
Eosinophils (% ± SD)	1.7 ± 2.1	1.8 ± 1.9
**BALF Biomarkers**		
KL-6 (U/mL)	868 ± 718	1489 ± 1090
SP-D (ng/mL)	332 ± 216	252 ± 207

Legend: SSc-ILD: Systemic Sclerosis Interstitial Lung Disease; lcSSc: Limited Cutaneous Systemic Sclerosis; dcSSc: Diffuse Cutaneous Systemic Sclerosis; PFT: Pulmonary Function TestsFVC: Forced Vital Capacity; FEV1: Forced Expiratory Volume in the 1st second. DLCO: Diffusion Lung CO; BAL: Bronchoalveolar Lavage.

**Table 2 pharmaceuticals-15-01584-t002:** Cellular percentages in BALF from SSc patients.

Paper	Neutrophils (% ± SD)	Lymphocytes (% ± SD)	Macrophages (% ± SD)	Eosinophils (% ± SD)
Cailes et al. [21]	3.00	6.00	85.00	1.00
De Santis et al. [23]	5.90 ± 10.50	4.30	6.10	1.10 ± 2.20
Kowal-Bielecka et al. [27]	5.40 ± 4.80	22.60 ± 13.70	70.20 ± 15.10	-
Moodley et al. [28]	10.30 ± 2.10	6.30 ± 2.20	79.60 ± 3.10	4.00 ± 1.00
Nagasawa et al. [29]	2.30 ± 2.90	17.20 ± 8.60	79.60 ± 9.30	1.00 ± 1.30
Prasse et al. [30]	9.00 ± 10.00	18.00 ± 10.00	69.00 ± 17.00	3.00 ± 2.00
Salaffi et al. [30]	4.10 ± 0.90	13.30 ± 2.10	80.90 ± 3.40	1.70 ± 0.70
Yilmaz et al. [34]	9.50 ± 12.30	18.80 ± 17.40	66.00 ± 19.50	-

Legend: Values are expressed as mean in percentage. SD: Standard deviation, when available, is in brackets.

**Table 3 pharmaceuticals-15-01584-t003:** Biomarkers values in BALF from SSc patients.

Cailes et al. [21]	FA-SSc (*n* = 45) % Patients	Molecular Level in BALF
MPO	0	0
Elastase		106 (3–263) ng·mL^−1^
Elastase/α_1_–antitrypsin complex		1.8(0.29–8.5) ng·mL^−1^
Collagenase		0–60 m units
Lactoferrin		0.17 (0.03–1.03) ng·mL^−1^
Kowal-Bielecka et al. [27]		
BTG	29.7% (11/37)	106.9 (15.2–229.0) IU/mL
PF4	21.6% (8/37)	35.5 (7.0–53.0) IU/mL

Legend: MPO: Myeloperoxidase; FA-SSc: Fibrosing alveolitis complicating systemic sclerosis; BTG: βthromboglobulin; PF4: Platelet factor 4.

## Data Availability

No new data were created or analyzed in this study. Data sharing is not applicable to this article.

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
