# Peer review of "The Role of Bronchoalveolar Lavage in Systemic Sclerosis Interstitial Lung Disease: A Systematic Literature Review"

_pharmaceuticals, 2022, doi:10.3390/ph15121584_

Round 1

Reviewer 1 Report

The systematic literature review is good manuscript. It is devoted to the problem of using BAL in SSc-ILD. A detailed overview is presented. The authors clearly described the article search criteria. Selected articles are discussed in detail. The authors concluded there are no clear criteria for BAL changing in the patients due to the lack of standards for performing BAL, as well as fast growth of modern analytical technologies. A gap in knowledge has been identified.

I have some criticisms of the article:
1. In Table 2, I see a strange percentage of macrophages in the paper by Clements et al [22]. Please check the data. The articles in Table 2 do not contain the data presented in the Table 2. How did you get them?
2. I suggest the authors to change the structure of the discussion:
BAL method
Patient’s related factors could influence BAL results
The role of alveolitis as a mirror of lung inflammation in ILD
The role of alveolitis in ILD prognostic evaluation
The role of BAL cytokine expression
Correlation between BAL and histopathological pattern
Correlation between BAL and HRCT features
The correlation of BAL and pulmonary functional tests
On my opinion, this structure is more logical. Perhaps, the authors will agree with me.
3. Discribe all abbreviations used in the abstract.
4. Unfortunately, I don't find Figure 1 in the paper or Supplementary Materials. So, I was not able to evaluate the criteria for excluding articles from this review.

Author Response

Thank you very much for your precious observations. We modified our manuscript according to your suggestion:

1. In Table 2, I see a strange percentage of macrophages in the paper by Clements et al [22]. Please check the data. The articles in Table 2 do not contain the data presented in the Table 2. How did you get them?

Point 1: We revised table 2

2. I suggest the authors to change the structure of the discussion:
BAL method
Patient’s related factors could influence BAL results
The role of alveolitis as a mirror of lung inflammation in ILD
The role of alveolitis in ILD prognostic evaluation
The role of BAL cytokine expression
Correlation between BAL and histopathological pattern
Correlation between BAL and HRCT features
The correlation of BAL and pulmonary functional tests
On my opinion, this structure is more logical. Perhaps, the authors will agree with me

Point 2: we modified the paragraphs' order, as you suggested 

3. Discribe all abbreviations used in the abstract.

Point 3: we specified abbreviation contained in the abstract

4. Unfortunately, I don't find Figure 1 in the paper or Supplementary Materials. So, I was not able to evaluate the criteria for excluding articles from this review.

Point 4: we inserted Figure 1 in supplementary materials.

Reviewer 2 Report

This review is devoted to the role of bronchoalveolar lavage in systemic sclerosis of interstitial lung disease. In my opinion, the review is well written and will be of interest to specialists in the field of the problem considered in it. In my opinion, the review can be published as presented. The only thing: in my opinion, splitting the review into chapters like "Introduction", "Materials and Methods", etc. not quite appropriate, because this is a review, not an experimental work.

Author Response

Thank you for your precious comments, 

we changed the paragraph "material and methods" to just "methods", in line with your suggestion

Reviewer 3 Report

The research article “The role of bronchoalveolar lavage in systemic sclerosis interstitial lung disease: a systematic literature review” presents a literature review on the role of BAL in evaluating the SSC-ILD. The article properly presents significant findings, although some improvements are needed. The article is suitable for publication in this journal after the MAJOR revision.

The authors should answer the following:

1.     The authors should check if the titles in line with names are allowed by the journal

2.     Line 61 – “databases were questioned for any relevant publications” – rephrase

3.     The references should be corrected according to the rules of the journal

4.     Figure 1 is mentioned in the Materials and Methods section but not presented in the manuscript

5.     The authors should list any computer programs used for the statistical analysis

6.     Additional explanations on the statistical tests used are needed

7.     The authors should cite some of the review papers that followed the same inclusion criteria or additionally explain why these criteria were selected (3. Results/Study selection)

8.     The authors should add more of the quantitative parameters in the Discussion section, as these statements are very general and unsupported by the actual data (4. Discussion, BAL method)

9.     Line 269 – “Most of the patients included in our SLR were affected by lcSSc, non-significant differences between lcSSc and dcSSc patients were found for what concerns BAL findings” – add actual quantitative data

10.  Lines 279 to 281 – add more quantitative data

11.  Lines 337 – 339 – the authors should verify that this discussion as it is the same as the Results

12.  Section 363 – the whole paragraph is without references and statistical parameters that are relevant for discussion

13.  The authors should add more quantitative data in the Conclusion to support these statements

Author Response

Thank you very much for your precious observations; according with you, we modified our manuscript.

In detail:

  1. The authors should check if the titles in line with names are allowed by the journal

Reply: We modified the title

  1. Line 61 – “databases were questioned for any relevant publications” – rephrase

Reply: We rephrased the indicated line

  1. The references should be corrected according to the rules of the journal

Reply: We corrected the references' format

  1. Figure 1 is mentioned in the Materials and Methods section but not presented in the manuscript

Reply: We added figure 1 in the manuscript and supplementary material

  1. The authors should list any computer programs used for the statistical analysis

Reply: We specified the software used for the descriptive statistical analysis

  1. Additional explanations on the statistical tests used are needed

Reply: We did not perform any statistical test because, as we stated in the manuscript, the included studies were to heterogeneous to allow a meta-analysis.

  1. The authors should cite some of the review papers that followed the same inclusion criteria or additionally explain why these criteria were selected (3. Results/Study selection)

Reply: We additionally explained the selection criteria. These criteria were chosen to select the most relevant articles and of the highest quality.

  1. The authors should add more of the quantitative parameters in the Discussion section, as these statements are very general and unsupported by the actual data (4. Discussion, BAL method)

Reply: We provided further numerical details

  1. Line 269 – “Most of the patients included in our SLR were affected by lcSSc, non-significant differences between lcSSc and dcSSc patients were found for what concerns BAL findings” – add actual quantitative data

Reply: We specified the reference of the signalled statement; in the cited reference, no quantitative data was given

  1. Lines 279 to 281 – add more quantitative data

Reply: As we stated in the manuscript, no univocal definition of alveolitis could be found in the selected studies and each paper shown a peculiar cut off for every cellular population. In table 1, we summarized by mean and standard deviation the differences between BAL with and without alveolitis, through the different studies

  1. Lines 337 – 339 – the authors should verify that this discussion as it is the same as the Results

Reply: We verified

  1. Section 363 – the whole paragraph is without references and statistical parameters that are relevant for discussion

Reply: We cited the only relevant paper about the argument (Bouros D.; Wells A.U.; Nicholson A.G.; Colby T.V.; Polychronopoulos V.; Pantelidis P.; Haslam P.L.; Vassilakis D.A.; Black C.M.; du Bois R.M. Histopathologic subsets of fibrosing alveolitis in patients with systemic sclerosis and their relationship to outcome. Am J Respir Crit Care Med. 2002;165(12):1581-6). We reported the significant statistical parameters

  1. The authors should add more quantitative data in the Conclusion to support these statements

Reply: We modified the Conclusion paragraph

Round 2

Reviewer 3 Report

The authors have answered all of the queries by the Reviewer.